# A Decision Support Tool to Optimize Selection of Head and Neck Cancer Patients for Proton Therapy

**DOI:** 10.3390/cancers14030681

**Published:** 2022-01-28

**Authors:** Makbule Tambas, Hans Paul van der Laan, Arjen van der Schaaf, Roel J. H. M. Steenbakkers, Johannes Albertus Langendijk

**Affiliations:** Department of Radiation Oncology, University Medical Center Groningen, University of Groningen, 9713 GZ Groningen, The Netherlands; h.p.van.der.laan@umcg.nl (H.P.v.d.L.); a.van.der.schaaf@umcg.nl (A.v.d.S.); r.steenbakkers@umcg.nl (R.J.H.M.S.); j.a.langendijk@umcg.nl (J.A.L.)

**Keywords:** proton therapy, head and neck cancer, plan comparison, IMPT, patient selection, model based selection, decision support tool, dose prediction

## Abstract

**Simple Summary:**

A decision support tool was developed to select head and neck cancer patients for proton therapy. The tool uses delineation data to predict expected toxicity risk reduction with proton therapy and can be used before a treatment plan is created. The positive predictive value of the tool is >90%. This tool significantly reduces delays in commencing treatment and avoid redundant photon vs. proton treatment plan comparison.

**Abstract:**

Selection of head and neck cancer (HNC) patients for proton therapy (PT) using plan comparison (VMAT vs. IMPT) for each patient is labor-intensive. Our aim was to develop a decision support tool to identify patients with high probability to qualify for PT, at a very early stage (immediately after delineation) to avoid delay in treatment initiation. A total of 151 HNC patients were included, of which 106 (70%) patients qualified for PT. Linear regression models for individual OARs were created to predict the D_mean_ to the OARs for VMAT and IMPT plans. The predictors were OAR volume percentages overlapping with target volumes. Then, actual and predicted plan comparison decisions were compared. Actual and predicted OAR D_mean_ (VMAT R^2^ = 0.953, IMPT R^2^ = 0.975) and NTCP values (VMAT R^2^ = 0.986, IMPT R^2^ = 0.992) were highly correlated. The sensitivity, specificity, PPV and NPV of the decision support tool were 64%, 87%, 92% and 51%, respectively. The expected toxicity reduction with IMPT can be predicted using only the delineation data. The probability of qualifying for PT is >90% when the tool indicates a positive outcome for PT. This tool will contribute significantly to a more effective selection of HNC patients for PT at a much earlier stage, reducing treatment delay.

## 1. Introduction

There is a remarkable increase in the number of head and neck cancer (HNC) patients treated with proton therapy (PT) worldwide [1,2,3]. Based on the 2019 data of the National Association for Proton Therapy survey, HNC patients constituted 14.2% of all patients treated in 28 PT centers in USA, compared to 5.8% in 2012 and showed the strongest increase in the number of patients treated with PT compared with other tumor sites [4]. A similar trend was also observed in European PT centers, with HNC being the most commonly treated indication following CNS tumors and comprising 15% of all adult patients treated with PT last year [5].

One of the main issues in PT application in HNC is how to select patient for PT who are likely to benefit most from PT in terms of toxicity risk reduction compared with photons. Given the fact that early treatment initiation is required for a better survival outcome in HNC [6,7,8], estimation of the expected benefit from PT as early as possible is clinically relevant. One way to determine the benefit from PT is to make a photon (volumetric-modulated arc therapy (VMAT)) vs. proton (intensity modulated proton therapy (IMPT)) treatment plan comparison and translate the differences in dose to organs-at-risk (OARs) (i.e., ΔDose), using normal tissue complication probability (NTCP) models, to expected toxicity risk differences (ΔNTCP) [9,10]. When using multiple NTCP-models, a ΔNTCP profile can be created that can be considered as a biomarker for the expected benefit of protons compared to photons. This called model-based selection has been used in the Netherlands since 2018 to select HNC patients for PT [9,11,12,13]. However, creating an in silico plan comparison for each patient is time consuming and may not be a feasible option for some centers. Centers that do not have a PT facility may refrain from consulting a PT center to check the suitability of a patient to be treated with PT, as this procedure may delay the initiation of treatment. They need to create a VMAT plan first, send it to a PT center combined with other patient data and then have to wait for an IMPT plan to be created and the results of a plan comparison being send back. It might result in treatment delays, especially for patients for whom a plan comparison result shows limited benefit from PT, in addition to the fee which may be asked by the PT center for the comparison that may not be covered by insurance companies.

Therefore, there is an unmet need for decision support tools which can estimate the expected benefit from PT without the need of a full plan comparison. The main advantages of such a tool would be to decrease delays in treatment initiation and to make efficient use of available resources. It may also allow for physicians to discuss the option for PT with their patients based on reliable estimates of the potential gain, which will thereby enhance shared decision making on a referral for PT, and whether this might be worthwhile [14,15,16,17].

The aim of this study was to develop a decision support tool to predict the toxicity risk reduction with PT (i.e., the ΔNTCP profile). The tool is to be used immediately after OAR and target volume delineation, but before performing any treatment planning. We aimed to investigate the use of delineation data only to predict both IMPT and VMAT OAR dose profiles. The decision support tool had to be straightforward, easy to implement in routine clinical practice while able to identify patients who are highly likely to gain from PT compared with regular photon treatment.

## 2. Materials and Methods

Our study comprised 151 patients treated with primary radiotherapy ± systemic treatment who were subjected to the model-based selection procedure between September 2019 and December 2020.

The OARs and target delineation was performed on the simulation CT according to the international consensus guidelines for CT-based delineation of OARs and targets in the head and neck region, using MRI and PET/CT imaging of the patients [18,19,20,21]. The radiotherapy schedule consisted of 54.25 Gy for PTV_5425 and 70 Gy for PTV_7000 in 35 fractions using a simultaneous integrated boost technique for both photons and protons (constant RBE 1.1). The characteristics of the VMAT and IMPT plans were discussed in detail in our previous study [22].

### 2.1. NTCP Models and *Δ*NTCP Thresholds

To test if patients qualified for PT, they were evaluated according to the updated Dutch National Indication Protocol for Proton Therapy, which includes four NTCP models, for Grade ≥ 2 xerostomia and dysphagia and Grade ≥ 3 xerostomia and dysphagia. In addition to baseline xerostomia and dysphagia complaints (Appendix A, Table A1), these NTCP models include the D_mean_ of eight OARs as predictors, including the oral cavity, bilateral submandibular and parotid glands and the pharyngeal constrictor muscle (PCM) superior, medius and inferior [23].

The ΔNTCP thresholds for PT selection were as follows: (1) ≥10% for Grade ≥ 2 toxicities, (2) ≥5% for Grade ≥ 3 toxicities, (3) ≥15% for the summed ΔNTCP (ΣΔNTCP) of Grade ≥ 2 toxicities (with a minimum of ≥5% for each) or (4) ≥5% for ΣΔNTCP of Grade ≥ 3 toxicities (with a minimum of ≥3.75% for each) [23].

### 2.2. IMPT and VMAT OAR D_mean_ Prediction

Overlap structures were created for each OAR for their part overlapping with PTV_7000 (OAR_in70) and for the OAR part outside the PTV_7000 but inside the PTV_5425 (OAR_in54_out70). Next, we expanded both of the PTVs with different margins including 3, 5, 7, 10 and 15 mm (with an estimated penumbra range of 0–15 mm) and created additional OAR overlap structures with these expanded PTVs (Figure 1).

In order to determine the optimal PTV expansions and corresponding OAR overlap structures for D_mean_ predictions, we created linear regression models for each OAR, where the endpoint was the D_mean_ of that OAR with either VMAT or IMPT. The predictors were the percentages of the OAR_in70 and OAR_in54_out70 volumes of that OARs. In total, 56 linear regression models were created (8 OARs x 6 different margins (0, 3, 5, 7, 10 and 15 mm) ×2 modalities (VMAT and IMPT)) to determine the most promising PTV expansion margin for OAR D_mean_ prediction. The performance of the models was evaluated by the goodness-of-fit R^2^ values (1 = perfect fit: predicted and actual values are equal). The margin that provided the model with the highest R^2^ value was selected for that OAR for VMAT and IMPT predictions.

### 2.3. The Proposed Decision Support Tool

Using the delineated OARs and target volumes and the overlap between them, the D_mean_ of the eight OARs in the NTCP models were predicted for VMAT and IMPT plans separately. Subsequently, NTCPs and ΔNTCP values were calculated. Based on the ΔNCTP thresholds mentioned above, the predicted plan comparison decision was determined for each patient (Figure 2).

#### Diagnostic Measures of the Decision Support Tool

The predicted and actual plan comparison outcomes were compared, and the sensitivity, specificity, positive and negative predictive value and accuracy of the proposed decision support tool were determined. In addition, the robustness of the tool within different patient subgroups (based on treatment initiation date and based on tumor location) were determined and compared with the 95% CI values of the decision support tool in the entire patient cohort for the following two reasons:-The quality of the plans can be improved when more experience is gained, as there is always a learning curve when a new treatment modality is implemented in a clinic (in this case IMPT) [24,25,26]. To account for that learning curve, patients were sorted based on their treatment initiation date and the population was divided into two subgroups. First, the initial 70 patients treated and second, the remaining 71 patients, who were treated more recently. Then, the diagnostic measures of the tool within these two subgroups were determined.-The D_mean_ of the OARs and the frequency of being selected for PT differ based on the primary tumor location, which may also impact the performance of the tool among patients with different tumor locations. In order to examine this, patients were divided into three different groups based on the primary tumor location, i.e., ‘pharynx’, ‘larynx’ and ‘others’. Subsequently, the diagnostic measures of the tool were determined within these three subgroups.

### 2.4. Statistical Analysis

A chi-squared test was used to examine the differences between groups by baseline categorical characteristics. For continuous variable comparisons, statistical tests were selected based on type (related and independent samples) and distribution (normal and non-normal) of the data. The differences in predicted OAR-doses and NTCP-values for VMAT and IMPT plans were compared using the Wilcoxon Signed Rank Test or the paired samples T-test, whichever appropriate. All statistical tests were two-sided and a *p*-value of ≤0.05 was considered statistically significant. Analyses were performed using the Statistical Package for Social Sciences (SPSS) for Windows, version 21.0 (SPSS Inc., Chicago, IL, USA).

## 3. Results

### 3.1. Patient and Selection for Proton Therapy

Of the 151 patients included in this study, 106 (70%) patients qualified for PT, while 45 (30%) patients did not and were treated with VMAT. Patients’ characteristics are shown in the Appendix A, Table A2.

For the different tumor locations, PT selection rates were different and the dominant NTCP model that triggered patient selection differed per location. In general, most patients were selected based on dysphagia related models or based on the ΣΔNTCP of grade ≥ 2 toxicities. However, laryngeal cancer patients were mainly selected based on xerostomia grade ≥ 2 models (Appendix A, Table A3).

### 3.2. VMAT and IMPT OAR D_mean_ Prediction Results

The R^2^ values of the 56 linear regression models created to predict VMAT and IMPT OAR D_mean_ are shown in Figure 3. The R^2^ values of the models for IMPT OAR D_mean_ predictions were generally higher than those for VMAT. The R^2^ values of the best performing linear regression models varied from 0.903 to 0.954 for VMAT and from 0.946 to 0.985 for IMPT.

#### Selected PTV Expansion Margins for D_mean_ Predictions

In general, 10 mm (for VMAT) and 5 mm (for IMPT) PTV expansion margins resulted in models with the highest R^2^ values (Figure 3). Eventually, for VMAT D_mean_ predictions, a 10 mm PTV expansion margin was selected for five of the eight OARs (PCM superior, PCM medius, PCM inferior and left and right submandibular gland), 7 mm PTV expansion margins were selected for two OARs (left and right parotid) and 15 mm for one OAR (oral cavity). On the other hand, for IMPT D_mean_ predictions, a 5 mm PTV expansion margin was selected for five of the eight OARs (PCM superior, PCM medius, right parotid and left and right submandibular glands) and 7 mm for three OARs (oral cavity, PCM inferior and left parotid gland). The coefficients of the selected models are given in the Appendix A, Table A4.

### 3.3. VMAT and IMPT NTCP Prediction Results

The predicted and actual NTCP values highly correlated with R^2^ values of 0.986 for VMAT and 0.992 for IMPT (Figure 4).

The residual values, i.e., the average ± SD differences between predicted and actual NTCP values for VMAT and IMPT, were 0.4% ± 2.6% and 0.1% ± 1.7% for Grade ≥ 2 dysphagia, 0.4% ± 2.0% and 0.2 ± 0.8% for Grade ≥ 3 dysphagia, 0.3 ± 2.2% and 0.0 ± 2.2% for Grade ≥ 2 xerostomia and 0.1 ± 0.9% and 0.0 ± 0.9% for Grade ≥ 3 xerostomia, respectively (Figure 5).

### 3.4. Diagnostic Measures of the Decision Support Tool

The sensitivity and specificity of the decision support tool were 64% (95% CI: 54–73) and 87% (95% CI: 73–95), respectively. The positive and negative predictive value and accuracy were 92% (95% CI: 84–96), 51% (95% CI: 44–58) and 71% (95% CI: 63–78), respectively. The post-hoc sensitivity analysis revealed that the diagnostic measures of the tool were within 95% CI limits among the first and second half of the patient population (Figure 6).

When the diagnostic measures of the decision support tool were evaluated based on the tumor location, a high variability was observed in terms of negative predictive value and sensitivity (Figure 6). In particular, sensitivity was low for the laryngeal tumors compared with other tumor locations.

When we compared the actual VMAT plan (instead of the predicted VMAT plan) with the predicted IMPT plan, the diagnostic measures of the decision support tool increased from 64 to 77% for sensitivity, 84 to 89% for specificity, 92 to 94% for positive predictive value, 51 to 63% for negative predictive value and 71 to 81% for accuracy.

## 4. Discussion

In this study, we developed a decision support tool to select patients for either VMAT or IMPT treatment based on the delineated OARs and PTVs before treatment planning. In our patient group, the positive predictive value of the tool was >90%, indicating that patients had a very high probability to be selected for PT in case of a positive decision support tool outcome. The negative predictive value of the tool was 51%, meaning that there is still approximately 50% probability that the patient will qualify for PT when the tool results a negative indication. The main advantage of such a tool is to identify HNC patients, at a very early stage of the preparation phase of radiotherapy, that are most likely to qualify for PT. The high positive predictive value of our tool can provide this information with a high level of confidence to both physicians and patients. In the Netherlands, in case of a positive outcome of the tool for PT, a plan comparison still has to be made as this is required to get reimbursement.

In our previous study, different versions of a pre-selection tool were proposed that required VMAT planning first or used a single overlap definition, only for PTV_5425 and which used the same margin for all OARs [27]. We now improved our models compared to our previous study. We predicted the ΔNTCP profile solely based on delineated structures without any treatment planning and determined the optimal PTV expansion margin for each OAR that could be used in conjunction with the percentage OAR overlap to predict both VMAT and IMPT OAR D_mean_, without the need to first perform VMAT planning.

To increase the level of confidence in the probability of being selected for PT, the NTCP profile of a VMAT plan created by the referring center could be compared with the predicted IMPT NTCP profile by the decision support tool for an even higher level of confidence, owing to the fact that the VMAT prediction is the main source of uncertainty and the correlation between the actual and predicted NTCP_IMPT_ values is relatively high (R^2^ = 0.992 vs. 0.946). Indeed, when we compared the actual VMAT plan (instead of the predicted VMAT plan) with the predicted IMPT plan, the performance of the decision support tool further improved.

Since the sensitivity and negative predictive value of the decision support tool is relatively low, institution-specific changes can be made to increase its sensitivity, i.e., to decrease false negative cases, depending on available resources, treatment capacity and preferences of that institution. In PT centers with sufficient resources where a high sensitivity is the priority, three different adjustments can be applied: (1) to rescale the predicted IMPT D_mean_ values using a uniform or OAR-specific rescaling factor, (2) to use either the lower (for IMPT) or upper (for VMAT) bound 95 CI% of the D_mean_ prediction models coefficients and (3) to decrease predicted IMPT D_mean_ values using the SD values of the residuals. This might be especially needed for patients with laryngeal cancer. The sensitivity of the decision support tool is lowest for this group, which can be explained by the relatively lower correlation between actual and predicted D_mean_ values for salivary glands compared with other OARs, given that laryngeal cancer patients qualify for PT mostly based on xerostomia-related ΔNTCP thresholds (Appendix A, Table A3). When these three proposed post-hoc adjustments were applied to the predicted D_mean_ values for salivary glands, the performance of the decision support tool for laryngeal cancer patients changed as follows: (1) when predicted IMPT D_mean_ was rescaled by 0.85, sensitivity increased from 25% to 63%, while specificity decreased from 100% to 50%, (2) when the lower bounds of the 95% CI of the coefficients were used to predict IMPT D_mean_, sensitivity was 57%, and specificity was 57% and (3) when predicted IMPT D_mean_ values were decreased by 1 SD of the residuals (3.1 Gy, see Figure 4), sensitivity was 81%, and specificity was 50%. If a sensitivity of 100% is the priority for a given institute, then a greater post-hoc adjustment can be applied, at the cost of plan comparison for more patients that would not eventually be selected for PT.

Automated planning combined with machine learning approaches can also be used to preselect patients for plan comparison based on delineation [28,29,30,31,32,33,34,35,36,37]. A recent study by Kouwenberg et al. investigated the potential of using automated planning in combination with machine learning to be used for preselection in 45 HNC patients who were subjected to model-based selection based on a previous version of the Dutch national indication protocol [37]. They compared the actual photon plan with an IMPT plan that was generated with non-clinical, fully-automated planning. Similar to our study, when the ΔNTCP thresholds were directly applied, it led to false negative and positive outcomes with an overall accuracy of 82%. To reach a sensitivity of 100% with minimal false positive results, the machine learning (Gaussian naïve Bayes classifier) was used to define the optimal decision boundary. It increased the sensitivity to 100% with an overall accuracy of 87%. Nevertheless, both their and our studies emphasize two main conclusions: (1) D_mean_ of the OARs can be predicted with a high accuracy either using automated planning or using a simpler approach of the OAR and PTV geometric relationship in terms of overlap; (2) using the predicted values directly leads to false positive and false negative results due to variation between manual and predicted doses, which can be overcome by using either advanced artificial intelligence methods as proposed by Kouwenberg et al. or using simpler post-hoc adjustments in the predicted values, as proposed in the current study.

The method proposed in the current study has some limitations, which are inherent to single center planning comparison study and, thus, the generalizability of the results in another settings. First, patients used in this study were treated with only two prescribed dose levels for the PTV (i.e., 54.25 and 70 Gy). Second, we used model-based optimization, in which the OARs were prioritized during optimization based on the NCTP gain per Gy increase in their D_mean_ values. Thus, it is very likely that the coefficients of the models used for the decision support tool are different in other centers where different dose schedules or other planning priorities are used [38]. Therefore, refitting the decision support tool models is warranted before clinical application in any center that would like to use it. Furthermore, the models need to be constantly updated as the expertise and optimization strategies evolve in time within a center and may differ from center to center. Moreover, technological developments in either photon or PT (e.g., dynamic arc proton therapy) over time may improve dose conformity and may jeopardize the performance of the tool, requiring further adjustments. Lastly, we currently present the diagnostic measures of the decision support tool based on the NTCP models and ΔNTCP thresholds as they are defined in the Dutch National Indication Protocol for Proton Therapy [23]. Different thresholds or protocols may be used to select HNC patients for PT in other countries.

## 5. Conclusions

We developed a decision support tool to select patients for PT that can be used before any treatment plan is created. The expected toxicity reduction with PT can be predicted using only the delineation data. The probability of qualifying for PT is >90% when the tool indicates a positive outcome for PT based on 5% and 10% NTCP reduction thresholds for grade ≥2 and ≥3 dysphagia and xerostomia, respectively. This tool can contribute significantly to identifying HNC patients at a much earlier stage that are highly like to benefit from PT. The tool avoids clinical workload, is cost effective and can be used without delaying treatment initiation.

## Figures and Tables

**Figure 1 cancers-14-00681-f001:**
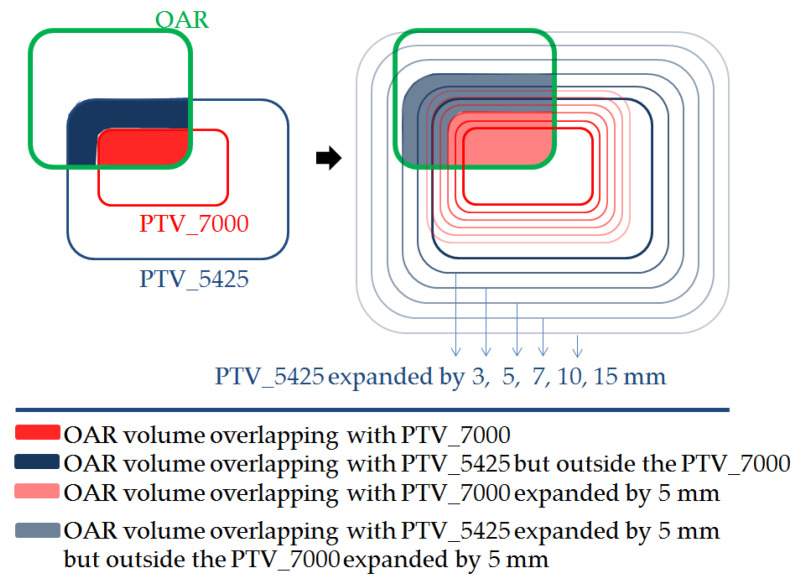
OAR volume overlapping with PTVs and PTVs expanded by different margins.

**Figure 2 cancers-14-00681-f002:**
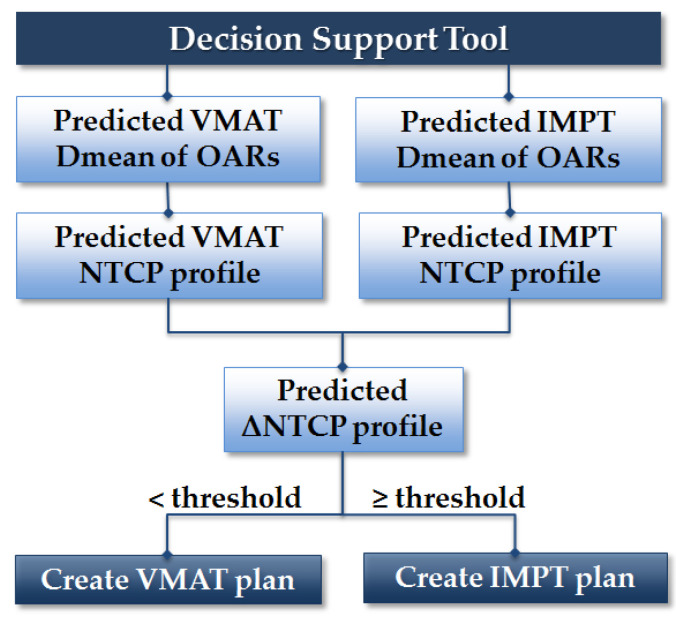
The workflow of the decision support tool.

**Figure 3 cancers-14-00681-f003:**
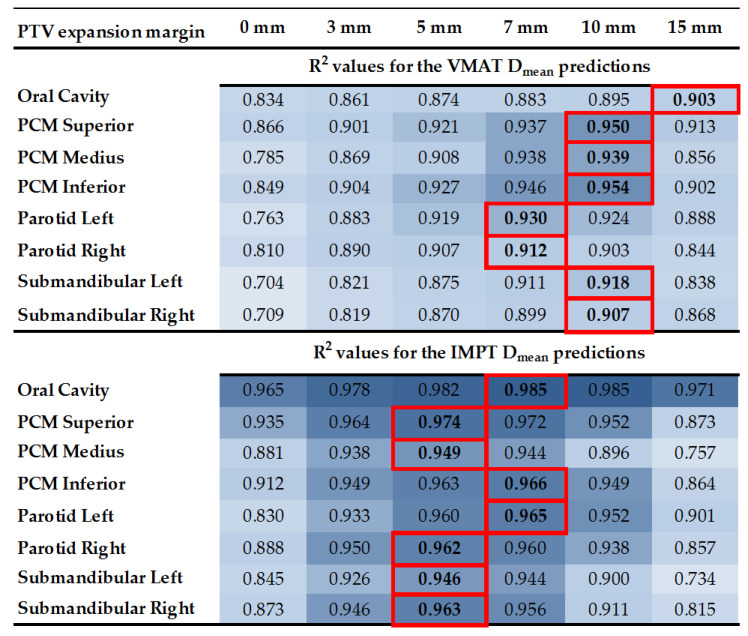
Assessment of the best performing PTV expansion margin to predict IMPT and VMAT OAR D_mean_. The R^2^ values are given with the highest R^2^ values for each OAR are outlined in red. The values become higher as the color of the cells gets darker.

**Figure 4 cancers-14-00681-f004:**
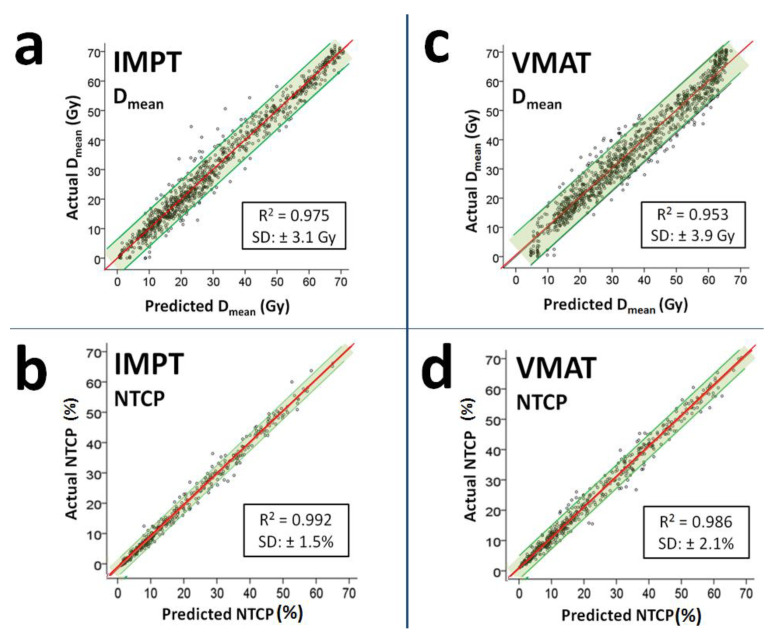
Predicted vs. actual values for eight OAR D_mean_ for IMPT (**a**) and VMAT (**c**); and predicted vs. actual four NTCP values for IMPT (**b**) and VMAT (**d**) shown in the same scatter plot. The red lines are the origins where predicted and actual values are equal to each other. The green areas indicate the 95% CIs for the individual predictions.

**Figure 5 cancers-14-00681-f005:**
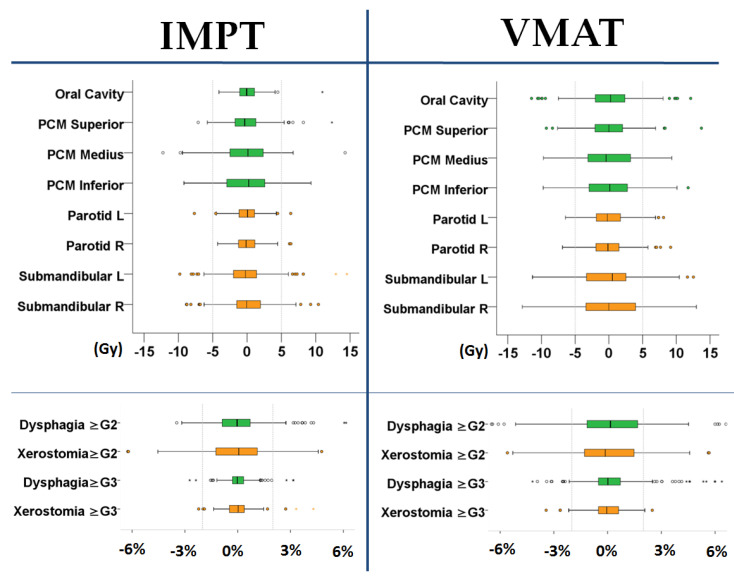
The boxplots of the residuals, i.e., the difference between predicted and actual OAR D_mean_ (**upper**) and NTCP values (**lower**) for IMPT and VMAT. Dysphagia- and xerostomia-related OARs and NTCPs are shown in green and orange, respectively. Dots in the figure represent outliers, i.e., values more than 1.5 interquartile range (IQR) but less than 3 IQR from the end of the box).

**Figure 6 cancers-14-00681-f006:**
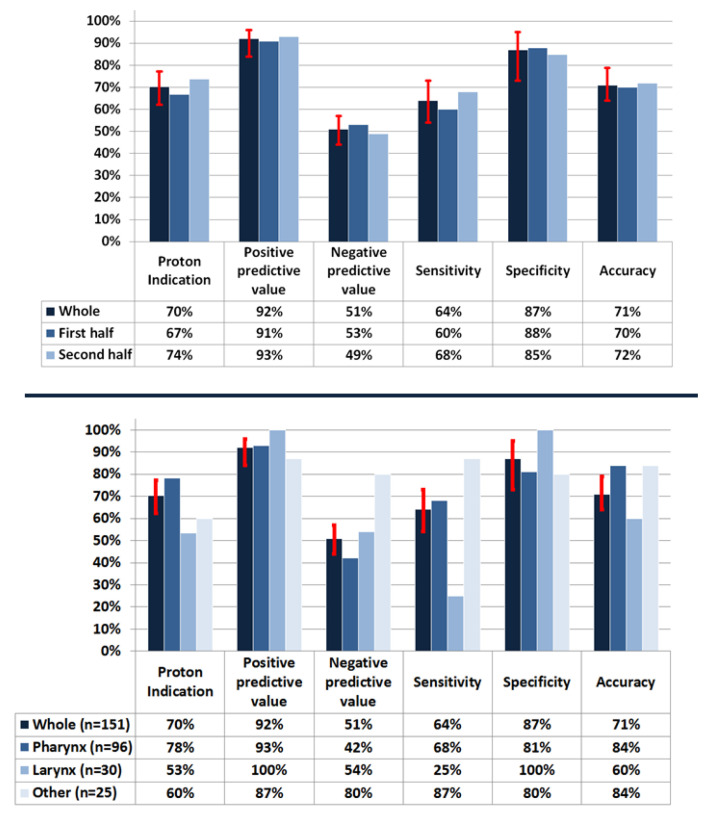
Variability of diagnostic measures across split cohort groups (**upper**) and across tumor locations (**lower**). The red bars on the columns of the whole patient cohort indicate 95% CI of the decision support tool.

## Data Availability

This study was based on the data derived from the prospective data registration program from department of Radiation Oncology of the UMCG. The data are part of a much larger project and will be used for a number of future research projects. Consequently, at present the data cannot be shared.

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
