# Peer review of "A Decision Support Tool to Optimize Selection of Head and Neck Cancer Patients for Proton Therapy"

_cancers, 2022, doi:10.3390/cancers14030681_

Round 1

Reviewer 1 Report

Dear Tambas et al.,

There are quite a number of articles of a very high quality from Your institution so it is always exciting to expect what comes next.

I attach the manuscript with some comments concerning mostly Materials and Methods part. 

Looking forward to read it again with the minor changes.

Best regards

Author Response

Response to Reviewer 1 Comments

Point 1: Line 80: I would believe that the problem in the last 2 sentences in this paragraph is first relevant when patient is receiving PT, not in relation to the choosing or planning the treatment modality. Is there a need to write about it ?

Response 1: We agree with the reviewer that the challenges stated here including patient travel and accomodation as well as reimbursement are more relevant after the patient is selected for proton therapy. Therefore, we deleted the two last sentences of this paragraph (please see: 1. Introduction / Paragraph 3).

Point 2: Line 117 and 120: You stated in the beginning that the tool is used before performing any treatment planning. Here you write in line both 117 and 120 about IMPT and VMAT. Maybe I am missing something, but it seems as OAR Dmeans are from IMPT and VMAT. Maybe you could phrase it differently?

Response 2: We understand reviewer’s comment and agree that it is confusing. Indeed, the decision support tool is intended to be used before any treatment planning is done. On the other hand, the preselection tool referred to in section 2.2. is different from the decision support tool developed in the present study. The preselection tool was developed in our previuos study, in which we developed different methods to create a preselection tool to select patients for IMPT planning once the VMAT plan has been created [1]. In the current study, we aimed to create a decision support tool that can be used in an earlier stage of RT treatment preparation (after delineation is completed and before any treatment planning) to predict the plan comparison result and expected toxicity risk reduction benefit from proton therapy.

Since it would be too complicated for the reader to understand the difference between preselection tool and decision support tool without reading our previous study and because the preselection tool is not the main focus of the current study, we have deleted this section from Materials and Methods to prevent any confusion and misunderstanding (Please see:. Materials and Methods).

In addition, the results section was also revised accordingly (please see: 3. Results / 3.1. Patient and selection for proton therapy)

Point 3: Line 160: Could you describe the importance of this? What consisted one and another groups of? Where there only two dates or how were these groups formed?

Response 3: We agree with the reviewer’s comment that the importance of analyzing the diagnostic measures of the decision support tool among different patient groups should have been described in more detail. Based on this comment, we revised this section and emphasized its significance and defined how the patient subgroups were formed as follows:

“The predicted and actual plan comparison outcomes were compared, and the sensitivity, specificity, positive and negative predictive value and accuracy of the proposed decision support tool were determined. In addition, the robustness of the tool within different patient subgroups (based on treatment initiation date and based on tumour location) were determined and compared with the 95% CI values of the decision support tool in the entire patient cohort for the following two reasons:

  • The quality of the plans can be improved when more experience is gained, as there is always a learning-curve when a new treatment modality is implemented in a clinic (in this case IMPT) [20-22]. To account for that learning curve, patients were sorted based on their treatment initiation date and the population was divided into two subgroups. First, the initial 70 patients treated and second, the remaining 71 patients, who were treated more recently. Then, the diagnostic measures of the tool within these two subgroups were determined.
  • The Dmean of the OARs and the frequency of being selected for PT differ based on the primary tumor location, which may also impact the performance of the tool among patients with different tumour locations. In order to examine this, patients were divided into three different groups based on the primary tumour location, i.e., ‘pharynx’, ‘larynx’ and ‘others’. Subsequently, the diagnostic measures of the tool were determined within these three subgroups.”

(please see: 2. Materials and Methods / 2.3. The proposed decision support tool / 2.3.1. Diagnostic measures of the decision support tool)

Point 4: Line 171: Does that mean 'below'?

Response 4: Yes, indeed it refers to “≤”. There was a typo and it has now been corrected (please see: 2. Materials and Methods / 2.4. Statistical Analysis).

References

[1] Tambas M, van der Laan HP, Rutgers W, van den Hoek JGM, Oldehinkel E, Meijer TWH, et al. Development of advanced preselection tools to reduce redundant plan comparisons in model-based selection of head and neck cancer patients for proton therapy. Radiother Oncol. 2021;160:61-8.

Reviewer 2 Report

Well-written manuscript that addresses an interesting but also very specific topic for radiation oncologists.  The study design is appropriate and the results are well presented. 

Two minor points  should be addressed.  

The authors do not provide any reference according to which guidelines/recommendations the target volume or the OAR contouring was carried out. 

Could the authors please state the period in which the patients were included in the study.

Author Response

Response to Reviewer 3 Comments

Well-written manuscript that addresses an interesting but also very specific topic for radiation oncologists.  The study design is appropriate and the results are well presented. 

Two minor points should be addressed.  

Point 1: The authors do not provide any reference according to which guidelines/recommendations the target volume or the OAR contouring was carried out. 

Response 1: We agree with the reviewer that guidelines used for OARs and target delineation should have been provided. Therefore, we have added a sentence about delineation and referred to the relevant guidelines in the Materials and Methods of the revised version of the manuscript as follows (please see: 2. Materials and Methods / Paragraph 2):

“The OARs and target delineation was performed on the simulation CT according to the international consensus guidelines for CT-based delineation of OARs and targets in the head and neck region, using MRI and PET/CT imaging of the patients [18-21].”

Point 2: Could the authors please state the period in which the patients were included in the study.

Response 2: Patients were included in the study between September 2019 and December 2020, which has already been stated in the manuscript as follows (please see: 2. Materials and Methods / paragraph 1):

Our study comprised 151 patients treated with primary radiotherapy ± systemic treatment who were subjected to the model-based selection procedure between September 2019 and December 2020.

References

  1. Brouwer, C.L., et al., CT-based delineation of organs at risk in the head and neck region: DAHANCA, EORTC, GORTEC, HKNPCSG, NCIC CTG, NCRI, NRG Oncology and TROG consensus guidelines. Radiother Oncol. 2015; 117: p. 83-90.
  2. Gregoire, V., et al., Delineation of the primary tumour Clinical Target Volumes (CTV-P) in laryngeal, hypopharyngeal, oropharyngeal and oral cavity squamous cell carcinoma: AIRO, CACA, DAHANCA, EORTC, GEORCC, GORTEC, HKNPCSG, HNCIG, IAG-KHT, LPRHHT, NCIC CTG, NRG Oncology, PHNS, SBRT, SOMERA, SRO, SSHNO, TROG consensus guidelines. Radiother Oncol. 2018; 126: p. 3-24.
  3. Lee, A.W., et al., International guideline for the delineation of the clinical target volumes (CTV) for nasopharyngeal carcinoma. Radiother Oncol. 2018 Jan;126(1): p. 25-36.
  4. Grégoire, V., et al., Delineation of the neck node levels for head and neck tumors: a 2013 update. DAHANCA, EORTC, HKNPCSG, NCIC CTG, NCRI, RTOG, TROG consensus guidelines. Radiother Oncol. 2014; 110: p. 172-181.